# Acetazolamide Therapy in Patients with Heart Failure: A Meta-Analysis

**DOI:** 10.3390/jcm8030349

**Published:** 2019-03-12

**Authors:** Janewit Wongboonsin, Charat Thongprayoon, Tarun Bathini, Patompong Ungprasert, Narothama Reddy Aeddula, Michael A. Mao, Wisit Cheungpasitporn

**Affiliations:** 1Department of Medicine, University of Minnesota, Minneapolis, MN 55455, USA; jwongboonsin@gmail.com; 2Division of Nephrology, Department of Internal Medicine, Faculty of Medicine, Siriraj Hospital, Mahidol University, Bangkok 10700, Thailand; 3Division of Nephrology and Hypertension, Mayo Clinic, Rochester, MN 55905, USA; charat.thongprayoon@gmail.com (C.T.); mao.michael@mayo.edu (M.A.M.); 4Department of Internal Medicine, University of Arizona, Tucson, AZ 85721, USA; tarunjacobb@gmail.com; 5Clinical Epidemiology Unit, Department of Research and Development, Faculty of Medicine, Siriraj Hospital, Mahidol University, Bangkok 10700, Thailand; p.ungprasert@gmail.com; 6Division of Nephrology, Department of Medicine, Deaconess Health System, Evansville, IN 47747, USA; dr.anreddy@gmail.com; 7Division of Nephrology, Department of Medicine, University of Mississippi Medical Center, Jackson, MS 39216, USA

**Keywords:** acetazolamide, diuretics, heart failure, sleep apnea, meta-analysis

## Abstract

Background and objectives: Fluid overload and central sleep apnea are highly prevalent in patients with heart failure (HF). We performed this meta-analysis to assess the effects of acetazolamide therapy on acid/base balance and apnea indexes. Methods: A literature search was conducted using EMBASE, MEDLINE, and Cochrane Database from inception through 18 November 2017 to identify studies evaluating the use of acetazolamide in HF. Study results were analyzed using a random effects model. The protocol for this systematic review is registered with PROSPERO (International Prospective Register of Systematic Reviews; no. CRD42017065401). Results: Nine studies (three randomized controlled trials and six cohort studies) with a total of 229 HF patients were enrolled. After acetazolamide treatment, there were significant decreases in serum pH (mean difference (MD) of −0.04 (95% CI, −0.06 to −0.02)), pCO_2_ (MD of −2.06 mmHg (95% CI, −3.60 to −0.53 mmHg)), and serum bicarbonate levels (MD of −6.42 mmol/L (95% CI, −10.05 to −2.79 mmol/L)). When compared to a placebo, acetazolamide significantly increased natriuresis (standardized mean difference (SMD) of 0.67 (95% CI, 0.08 to 1.27)), and decreased the apnea-hypopnea index (AHI) (SMD of −1.06 (95% CI, −1.75 to −0.36)) and central apnea index (CAI) (SMD of −1.10 (95% CI, −1.80 to −0.40)). Egger’s regression asymmetry tests revealed no publication bias with *p* = 0.20, 0.75 and 0.59 for analysis of the changes in pH, pCO_2_, and serum bicarbonate levels with use of acetazolamide in HF patients. Conclusion: Our study demonstrates significant reduction in serum pH, increase in natriuresis, and improvements in apnea indexes with use of acetazolamide among HF patients.

## 1. Introduction

Heart failure affects 1–2% of the world population with a higher crude prevalence in high-income countries [1]; however, its economic burden has grown globally. Fluid management is one of the therapeutic keystones in assuring symptom and hospital admission control. Despite the ongoing development of advanced therapies, heart failure patients are facing new challenges due to a greater prevalence of advanced concurrent chronic kidney disease (CKD) and diuretic resistance [2]. While loop diuretics and thiazide diuretics are the primary diuretic therapy choice, other nephron tubular sites for diuretic blockage have not been commonly utilized and could serve as another avenue for intervention. 

Acetazolamide is a carbonic anhydrase inhibitor that blocks proximal tubular absorption of sodium. Acetazolamide was first used in humans as a new diuretic with potential to treat congestive heart failure [3] after it was found to be more potent and safer than sulfanilamide diuretic. Several case studies in the 1950s that utilized acetazolamide in heart failure patients demonstrated successful decongestive profiles [4,5,6]. Its use had been declining with the discovery of loop diuretics, which are perceived as more potent [7]. However, there were new pieces of evidence showing that the ceiling effect of acetazolamide was due to a compensatory increase of distal tubular Na-Cl co-transporter mediated by decreased expression of pendrin, making dual use of acetazolamide and thiazide diuretics a promising therapy for diuretic resistance [8,9]. Recently, Imiela et al. performed a randomized control trial on the effect of acetazolamide on heart failure, showing a positive effect on diuretic augmentation [10]. There are ongoing clinical trials that specifically investigate the effect of acetazolamide on volume overload state; however, the results are still pending. Therefore, a systematic review could provide a comprehensive background and understanding of current evidence in this clinical scenario.

Acetazolamide has also been routinely used in altitude medicine [11] to prevent and treat fluid retention and pulmonary edema by increasing diuresis, improving oxygenation and decreasing periodic breathing patterns [12]. Patients with central sleep apnea (CSA) have demonstrated better control of the apnea–hypopnea index (AHI) when treated with acetazolamide; however, its application in heart failure patients with sleep disorders has not yet been implemented as common practice due to concerns of urinary potassium wasting and the resulting potential arrhythmias [13]. As central sleep disorder is a relatively common comorbidity in heart failure patients, with a reported rate of up to 30–50% in heart failure patients experiencing reduced left ventricular ejection fraction, acetazolamide could be an additional therapeutic option that would not only help with the volume aspect of heart failure but also the improvement of respiratory indices.

We performed a systematic review and meta-analysis to measure the overall potency of acetazolamide in heart failure patients with a focus on acid/base alterations and alleviation of sleep apnea indices.

## 2. Methods

### 2.1. Search Strategy and Literature Review

The protocol for this systematic review is registered with PROSPERO (International Prospective Register of Systematic Reviews; no. CRD42017065401). A systematic literature search of EMBASE (1988 to 18th November 2017), MEDLINE (1946 to 18th November 2017), and the Cochrane Database of Systematic Reviews (database inception to 18th November 2017) was performed to identify studies evaluating the use of acetazolamide in HF patients. The systematic literature review was performed independently by two investigators (J.W. and C.T.) using the keywords and medical subject heading (MeSH) terms of “acetazolamide” or “diamox”, and “heart failure” (provided in online Appendix A). Additionally, a manual search for conceivably relevant articles using references of the included articles was also performed. This study was conducted in accordance with the STROBE (Strengthening the Reporting of Observational Studies in Epidemiology) [14] and the PRISMA (Preferred Reporting Items for Systematic Reviews and Meta-Analysis) statements [15] as described in online Appendix A.

### 2.2. Selection Criteria

Eligible studies consisted of clinical trials or observational studies (cohort, case-control, or cross-sectional studies) that assessed the effects of acetazolamide therapy on 1) acid/base balance and 2) apnea indexes. The studies were included only if they provided data allowing for the calculation of mean differences (MDs), standardized mean differences (SMDs), relative risks, or hazard ratios with 95% confidence intervals (CI). Retrieved articles were individually reviewed for eligibility by the two investigators (J.W. and C.T.). Discrepancies were resolved by mutual consensus. The quality of each included study was quantified via the Cochrane risk of bias tool [16]. The Newcastle-Ottawa quality assessment scale was utilized to appraise the quality of study for observational studies [17].

### 2.3. Data Abstraction

A structured data collecting form was utilized to derive the following information from each study: name of the first author, title, year of the study, publication year, country where the study was conducted, demographic and characteristic data of heart failure patients, methods used to identify heart failure, acetazolamide regimen and dosages, data on fluid and electrolytes, apnea indexes, and adjusted effect estimates with 95% CI. 

### 2.4. Statistical Analysis

Comprehensive Meta-Analysis software (version 3.3.070; Biostat Inc, Englewood, New Jersey, USA) was used for the analyses. Adjusted point estimates from each included study were consolidated by the generic inverse variance approach of DerSimonian and Laird, which designated the weight of each included study based on its variance [18]. Given the likelihood of between-study variance, we used a random-effect model rather than a fixed-effect model. Cochran’s Q test with I^2^ statistics were applied to assess the between-study heterogeneity. An I^2^ value of 0% to 25% indicates insignificant heterogeneity, 26% to 50% low heterogeneity, 51% to 75% moderate heterogeneity and 76% to 100% high heterogeneity [16,19]. The Egger test was used to assess the presence of publication bias [20].

## 3. Results

A total of 614 potentially eligible articles were identified using our search strategy. After excluding 571 articles that did not fulfill inclusion criteria on the basis of type of article, study design, population, and outcome of interest, 43 articles were left for full-length review (Appendix A). Thirty-four of the final 43 articles were additionally excluded from full-length review, as shown in Figure 1. Thus, the final analysis included nine articles consisting of three randomized controlled trials and six cohort studies with a total of 229 HF patients. The literature retrieval, review, and selection process are demonstrated in Figure 1. 

The characteristics and quality assessment of the included studies are presented in Table 1 and Table 2. Table 1 describes the main characteristics of experimental studies that utilized acetazolamide as a treatment agent in patients with heart failure. The two studies from Javaheri et al. [21,22] used acetazolamide as an experimental drug and monitored several sleep disorder parameters. Both studies shared similar methodologies. The third study by Imiela et al. [10] was one of the first randomized controlled trials that was dedicated to heart failure outcomes including urine output and clinical measurements. Table 2 summarizes important descriptions of observational studies that focused on acetazolamide in various settings of heart failure patients. It is of note that three out of six were performed prior to the era of evidence-based medicine; therefore, case definition and measured parameters may deviate from the other studies. Amongst the other three studies that were performed more recently, all share similar quality of case selection and ascertainment of exposure (diuretics).

### 3.1. Effects of Acetazolamide on Acid/Base Balance in HF Patients

After acetazolamide treatment, there were significant decreases in serum pH (mean difference (MD) of −0.04 (95% CI, −0.06 to −0.02), I^2^ = 65%, Figure 2A), pCO_2_ (MD of −2.06 mmHg (95% CI, −3.60 to −0.53 mmHg), I^2^ = 0%, Figure 2B) and serum bicarbonate levels (MD of −6.42 mmol/L (95% CI, −10.05 to −2.79 mmol/L), I^2^ = 95%, Figure 2C). This finding confirmed the expected effects of acetazolamide use in our population of interest. The changes of pCO_2_ appeared to be less robust than serum bicarbonate level as reflected by a wider confidence interval in each study. This may be due to variation in the time of blood specimen collection, allowing more variation in the respiratory compensation in response to induced metabolic acidosis. When compared to a placebo, acetazolamide significantly increased natriuresis (standardized mean difference (SMD) of 0.67 (95% CI, 0.08 to 1.27)), I^2^ = 0%, Figure 3). 

### 3.2. Effects of Acetazolamide on Apnea Indexes

When compared to a placebo, acetazolamide treatment significantly reduced the apnea-hypopnea index (AHI) (SMD of −1.06, (95% CI, −1.75 to −0.36), I^2^ = 0%, Figure 4A) and central apnea index (CAI) (SMD of −1.10, (95% CI, −1.80 to −0.40)), I^2^ = 0% Figure 4B), respectively. It is important to note that both of the studies included in this analysis came from the same investigator with a very similar study protocol used in each study. As measurement of sleep parameters required an elaborate setting to be quantified, other studies did not contain the data required to be included in the meta-analysis.

### 3.3. Sensitivity Analysis

Sensitivity analysis was performed by excluding one study at a time to investigate the effect of each study on the pooled MD and SMD for each outcome assessed. The pooled effect estimate from the sensitivity analysis remained essentially unchanged. 

### 3.4. Evaluation for Publication Bias

We found no publication bias as assessed by the Egger’s regression asymmetry test with *p* = 0.20, 0.75 and 0.59 for analysis of the changes of serum pH, pCO_2_, and serum bicarbonate levels with use of acetazolamide in HF patients, respectively. Since a limited number of included studies evaluated the changes of AHI and CAI, the power of the test is too low to evaluate the publication bias. 

## 4. Discussion

In this meta-analysis, we consolidated the effects of using acetazolamide in populations of heart failure patients. By inducing bicarbonaturia at the level of the proximal tubules, acetazolamide led to a modest decrease in serum bicarbonate (MD of 6.42 mmol) without a considerable disruption of pCO_2_ and pH in this patient population. This suggests that an acetazolamide-induced metabolic acidosis could assist respiratory compensation even in patients with heart failure. There is a reasonable need for caution towards medications that can cause an abrupt change in pH due to their potential to cause an arrhythmogenic state. Our study, however, showed that acetazolamide use in heart failure only caused minor changes in serum pH (MD of 0.04), making it a reasonable agent to use in patients who have developed contraction alkalosis from diuresis. In addition, acetazolamide increased natriuresis compared to a placebo, thus supporting prior evidence that showed its use as an adjunct to other diuretics [8,9,10]. One important note for the pharmacodynamic profile of acetazolamide was the development of tolerance after consecutive days of daily administration [29], which resulted in decreased natriuresis and potassium excretion after 48–72 h. The authors suggested limiting acetazolamide use to no more than three to four days a week or for less than two consecutive days at a time in order to sustain its desired diuretic properties. The mechanism of this phenomenon was unclear and it was unknown whether the effect of acid-base changes may also be affected, suggesting a topic for further experimental research. 

Acetazolamide was also shown to improve apnea–hypopnea index and central apnea index in our study. This finding expanded the potential use of acetazolamide in heart failure patients, as sleep apnea is a common comorbidity that is often exacerbated with heart failure decompensation [30]. While controversy existed around the risk or benefit of adaptive servo-ventilation (ASV) use in the heart failure population, continuous positive airway pressure (CPAP) is well accepted as a modality of treatment for obstructive sleep apnea (OSA) in the heart failure population. As compliance with CPAP use remains a substantial barrier to effective treatment of sleep apnea, acetazolamide could potentially serve as an alternative therapeutic modality for this patient population. The mechanism of acetazolamide effect on sleep indices is still unclear. Prior studies have found evidence that acetazolamide paradoxically improved apnea indices despite augmenting hypercapnic ventilatory response, (HCVR) which was thought to be the main pathophysiology of central sleep apnea [22]. Authors have proposed other mechanisms including changes in O2 and CO2 chemoreceptor sensitivity as the possible explanation of how acetazolamide led to more favorable apnea indices. It is important to note that in two of the studies that we analyzed, the reduction of AHI was not associated with increased diuresis or an improved cardiac status, suggesting that the effect of acetazolamide in optimizing sleep indices may be independent of the patient’s volume status. Future studies exploring the effect of acetazolamide on sleep indices after diuresis has achieved the desired volume status in heart failure patients, and whether the independent beneficial effect persists, will be important. When compared to the non-heart failure central sleep apnea patients, several studies addressed the effect of acetazolamide on CSA and OSA. There was a general trend of acetazolamide improving the CSA sleep parameter and sleep-related symptoms, but there conflicting evidence on the effect of OSA sleep parameter. A recent publication by Aurora et al. [31] proposed acetazolamide as an option of CSA for both the heart failure and the non-heart failure patient but acknowledged the potential side effect of the medication. Given the complexity of non-heart failure CSA patients with persistent of CO_2_ receptor reset [32], ongoing research exploring the impact of acetazolamide in a CHF patient may benefit from additional outcome adjudication after acetazolamide therapy was interrupted.

Potential deleterious effects of acetazolamide include metabolic acidosis, which could transiently increase the sympathetic response of heart failure [21]. It is also possible that hyperventilation as a result of acetazolamide use could lead to increased fatigability of respiratory muscles during the treatment period. Several early studies of acetazolamide reported side effects such as nausea, vomiting, paresthesia, dizziness, and muscle cramps. These symptoms were attributed to electrolyte and acid-base shifts that occurred either from diuresis or were a direct effect of the carbonic anhydrase inhibition on nerve cells [33]. Potassium monitoring and appropriate supplementation will be critical in the effective use of acetazolamide, as reflected in multiple prior studies that included potassium supplementation in the study protocol. In the study from Javaheri 2006 which utilized a slightly higher dose of acetazolamide, one participant reported shortness of breath during the study. The author proposed that the effect was due to the dosage of the medication being higher than what would be used in clinical practice. None of the patients in the study reported cases of paresthesia.

There are several limitations to this meta-analysis. First, the quality of the observational studies was limited as many were performed before the era of modern evidence-based medicine. However, the three selected studies described their protocol meticulously, and by further utilizing the same patients for both the control and exposure population, it resembled a time series methodology. Second, the three cohort studies included were very well-designed studies but were not primarily looking at the effects of acetazolamide on the outcome of our interest; this would introduce potential bias by analyzing previous reported secondary data. On the other hand, the data from the three randomized control trials were of high quality given its sole use of acetazolamide as the main intervention. Only one study did not provide blinding for acetazolamide use, which likely could have led to detection bias in the subjective outcome (dyspnea scale) of the study. However, our pooled data of urinary sodium excretion should be minimally affected. In addition, we performed an additional search of the literature to cover recently published articles up to the time of publication. We identified seven other items [34,35,36,37,38,39,40], two of which would qualify for our screening criteria. However, both of the articles did not contain our outcome of interest even though part of the studies involved the usage of acetazolamide. Thus, future large-scale clinical trials are required to confirm these effects of acetazolamide therapy on acid/base balance and apnea indexes.

In summary, our systematic review and meta-analysis suggest that acetazolamide may have a potential beneficial effect in select heart failure patients by reducing serum bicarbonate level and augmenting natriuresis with only slight changes in serum pH and pCO_2_. Acetazolamide also helped to reduce sleep apnea indices, expanding its potential use for sleep apnea in the heart failure population. Further studies to explore the mechanism, safety, and beneficial effects of acetazolamide when used in conjunction with other diuretics would be an important next step prior to its widespread adoption in the clinician’s armamentarium. 

## Figures and Tables

**Figure 1 jcm-08-00349-f001:**
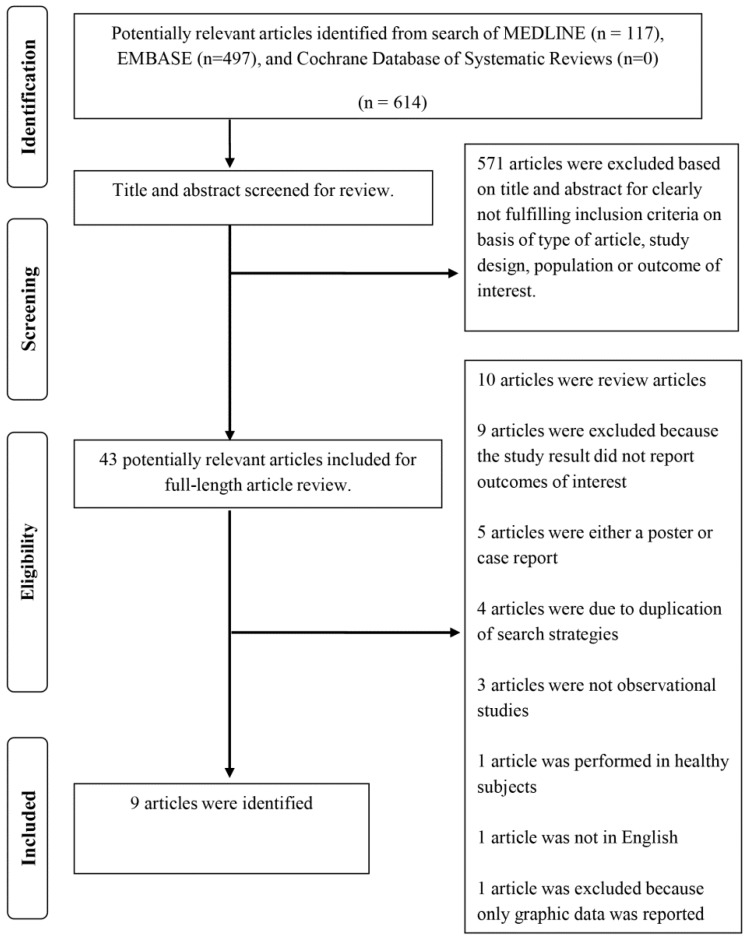
Outline of our search methodology.

**Figure 2 jcm-08-00349-f002:**
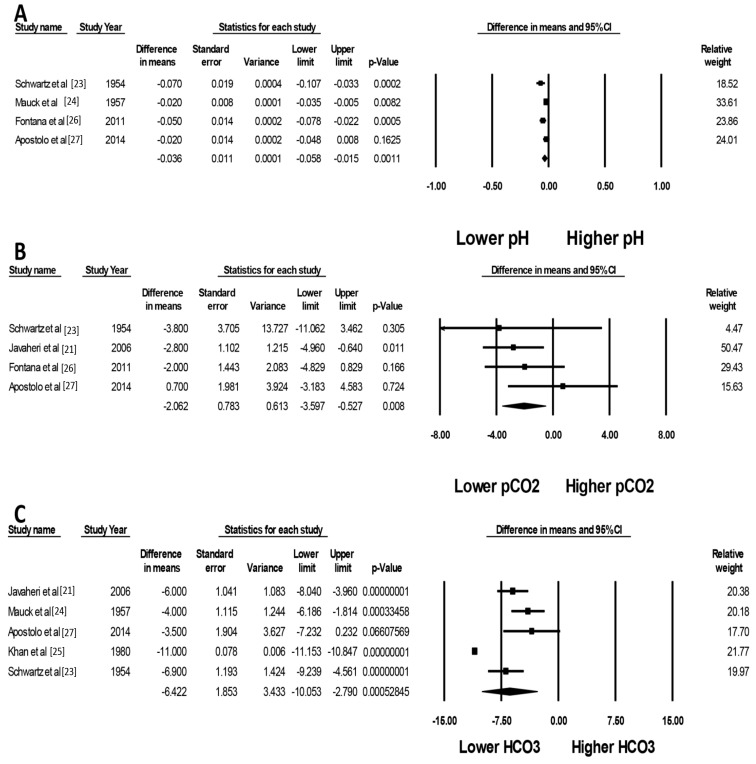
Forest plots of the included studies assessing the effects of acetazolamide on (**A**) pH; (**B**) pCO_2_; and (**C**) bicarbonate. A diamond data marker represents the overall rate from each included study (square data marker) and 95% confidence interval (CI).

**Figure 3 jcm-08-00349-f003:**
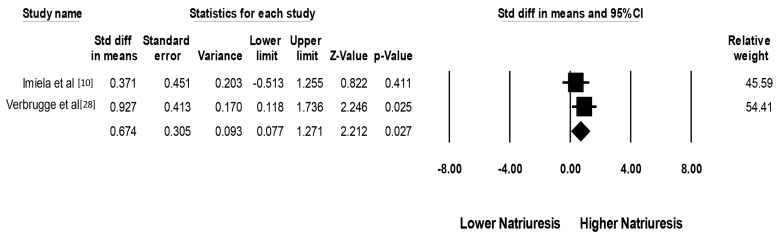
Forest plot of the included studies assessing effects of acetazolamide on natriuresis, when compared to control. A diamond data marker represents the overall rate from each included study (square data marker) and 95% CI.

**Figure 4 jcm-08-00349-f004:**
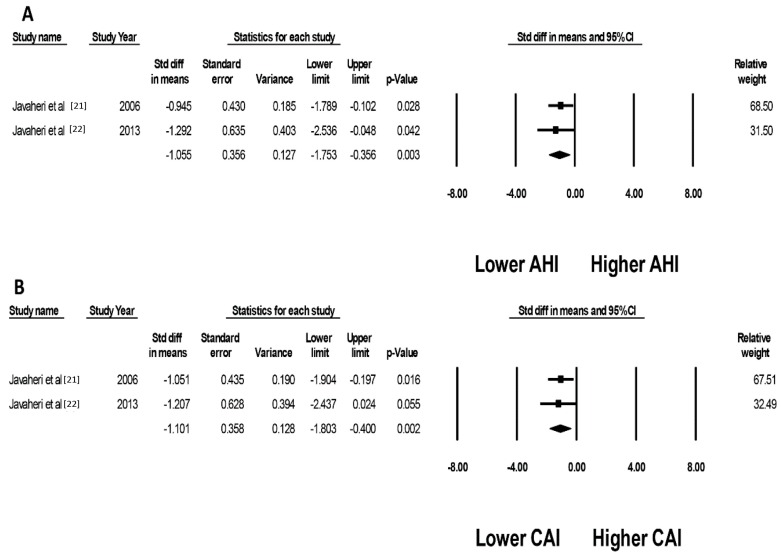
Forest plots of the included studies evaluating effects of acetazolamide on (**A**) apnea-hypopnea index (AHI); and (**B**) central apnea index (CAI). A diamond data marker represents the overall rate from each included study (square data marker) and 95% CI.

**Table 1 jcm-08-00349-t001:** Main characteristics of the RCTs included in the meta-analysis of acetazolamide therapy in heart failure patients.

	Study (Author, Year)	Javaheri, 2006 [21]	Javaheri, 2014 [22]	Imiela, 2017 [10]
Method	Study design	RCT, double-blind cross-over placebo and acetazolamide	RCT, double-blind cross-over placebo and acetazolamide	RCT, single-center, unblinded
	Total number	12	6	20
	Patient sample	Stable systolic heart failure	Stable systolic heart failure	Patient hospitalized with chronic heart failure exacerbation
	CHF definition	Systolic HF (EF < 35%) with Cheyne-Stokes breathing and AHI > 15/h	Systolic heart failure and CSA (AHI > 15/h)	EF < 50%, signs of volume overload
	Acetazolamide dosing	Acetazolamide 3.5–4 mg/kg + KCL 60 meq	Acetazolamide 3.5–4 mg/kg + KCL 60 meq	Once daily (dose-adjusted to body weight, range from 250 mg to 500 mg) as add-on diuretics to furosemide
	Description	RCT about acetazolmide but designed for sleep disorder	RCT about acetazolmide but designed for sleep disorder	RCT dedicated to acetazolamide
Cochrane risk of bias	Selection	Low	Low	Low
	Performance	Low	Low	High
	Detection	Low	Low	High
	Attrition	Low	Low	Low
	Reporting	Low	Low	Low
	Other	Low	Low	Low

Abbreviations: RCT, randomized controlled trial; CHF, congestive heart failure; EF, ejection fraction; AHI, apnea-hypopnea index; CSA, central sleep apnea; KCL, potassium chloride.

**Table 2 jcm-08-00349-t002:** Main characteristics of cohort studies included in meta-analysis of acetazolamide therapy in heart failure patients.

	Study (Author, Year)	Schwartz, 1955 [23]	Mauck, 1957 [24]	Khan, 1980 [25]	Fontana, 2011 [26]	Apostolo, 2014 [27]	Verbrugge, 2015 [28]
**Method**	Study design	Cohort, time series	Cohort, time series	Cohort, time series	Cohort	Cohort	Cohort
	Total number	17	14	74	12	20	54
	Patient sample	Severe CHF with cor pulmonale	Stable CHF	Hospitalized decompensated CHF	CHF with periodic breathing	CHF with periodic breathing	Hospitalized decompensated CHF
	CHF definition	Clinical diagnosis with severe pulmonary disease, with ECG or Xray change of right heart enlargement or strain	Fully digitalized ambulatory CHF on 1 gm diet, mostly edema free. Wash out from other diuretics for two weeks	Patients diagnosed with CHF that failed treatment with furosemide, spironolactone and salt restriction	EF < 50%, with AHI > 15	EF < 40%, with periodic breathing during exercise	At least three signs of volume overload, EF < 45%
	Acetazolamide dosing	1–1.5 gm over 24 h	250 mg qday	250 mg qid (1 g per day), used as a replacement of furosemide	250 mg po bid for 4 days	500 mg IV and prolonged 2 gm over 24 h	250 mg qday
	Description	Using acetazolamide to treat patient with cor pulmonale and respiratory acidosis	Studying effect of acetazolamide in ambulatory CHF, aiming at electrolyte values during therapy	Using acetazolamide instead of furosemide for diuresis	Measure breathing parameters with ABG analysis before and after acetazolamide treatment	Measured respiratory pattern before and after acetazolamide treatment	Cohort finding predictor of natriuresis
**Quality assessment (Newcastle-Ottawa scale)**	Selection	2	2	2 - Independent validation - Obvious series of cases	3 - Using cohort of CHF - Non-exposed cohort is self prior to acetazolamide - Secure record	3 - Using cohort of CHF - Non-exposed cohort is self prior to acetazolamide - secure record	3 - True representation of CHF - Same community of control - Secure record
	Comparability	1	1	1 - Study controls by using prior-self prior to treatment with acetazolamide	1 - Study controls by using prior-self prior to treatment with acetazolamide	1 - Study controls by using prior-self prior to treatment with acetazolamide	2 - Using multivariate analysis and still showed acetazolamide as important factor
	Outcome	1	1	1 -Secure record (hospital)	NA	NA	NA
	Exposure	NA	NA	NA	2 - Record linkage - Long follow up	2 - Record linkage - Long follow up	3 -Record linkage - Long follow up - Complete follow up of all subjects

Abbreviations: CHF, congestive heart failure; EF, ejection fraction; AHI, apnea–hypopnea index; ECG, electrocardiogram; IV, intravenous; ABG, arterial blood gas; NA, not applicable.

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
