# Peer review of "Acetazolamide Therapy in Patients with Heart Failure: A Meta-Analysis"

_jcm, 2019, doi:10.3390/jcm8030349_

Round 1
Reviewer 1 Report
interesting paper,
the paper is clear and well written.
there are 2 tables and 3 figures in the support of the results.
The methods are adequately described and the result are clearly presented.
The results and conclusion are very interesting.
English language and style are fine.
Author Response
Response: We thank you for reviewing our manuscript and for your critical evaluation. We really appreciated your input and appreciate you found our manuscript interesting and well written. We appreciated your time reviewing our manuscript
All authors thank the Editors and reviewers for their valuable suggestions. The manuscript has been improved considerably by the suggested revisions!

Reviewer 2 Report
The paper in question is a meta-analysis on the use of acetazolamide in heart failure patients, investigating its use both as a diuretic and as a sleep regulatory agent.
The study is well conducted from the procedural point of view. The technique used in the meta-analysis is correct, responds to widely accepted criteria and reaches well-documented conclusions.
However, I would like to highlight some critical points:
1- not all of the 9 articles included in the meta-analysis are cited in the bibliography; only two are mentioned, one of which, if I understand correctly, with the wrong year (2013 in the table, 2014 in references);
2- Acetazolamide is an "old" drug, known since the 50s, which was widely used at the beginning, then it was replaced by loop diuretics and recently it is back in vogue as support / augmentation diuretic agent in cases of more refractory decompensation; data on the latter recent use are well known and accepted;
3- It follows that the need for a meta-analysis is a bit "in the background" and this is even more true considering the low number of studies selected and included in the analysis (9 studies, 3 of them RCT, 2 on sleep disorder and just 1 on diuretic effect);
4- Authors themselves underline a limitation related to the cohort studies, so practically the meta-analysis on the diuretic effect is based just on a single paper;
5- The uniformity of the results of the studies is confirmed by the pooled effect estimate from the sensitivity analysis that remained essentially unchanged and by the absence of any publication bias;
6- The study dates back a little more than a year ago (2017) but in 2018 another 6 studies were published, one of which was randomized (ADVOR, still recruiting): perhaps it was also worth including them in the analysis.
Meta-analysis is a statistical procedure for combining data from multiple studies. When the treatment effect is consistent from one study to the next, meta-analysis can be used to identify this common effect. When the effect varies from one study to the next, meta-analysis may be used to identify the reason for the variation. When the utility of an intervention or the validity of a hypothesis cannot be based on the results of a single study, meta-analysis is needed to synthesize data across studies. Narrative review is largely subjective and become unuseful when there are more than a few studies involved, mainly if the studies come to different conclusions. Meta-analysis, by contrast, applies objective formulas and can be used with any number of studies, mainly if the studies come to different conclusions.
There are examples of extensive meta-analyses that help a lot in the overall view of a problem, because it would be impossible for a single person to analyze all the articles related to it and draw general weighted conclusions.
(for example: Gong B, J Cardiothorac Vasc Anesth 2015 Dec 29(6) on levosimendan, collecting 5349 pts from 25 RCTs; Burnett H, Circulation Heart Failure 2017 Jan 13 on drug for HF collecting 57 RCTs studies).
In a case like this there is a low number of articles that all agree with each other, reporting, as confirmed by the Authors themselves, the same conclusions. In particular, the diuretic effect is analysed just by means of a single, already known RCT.
Therefore I believe that a meta-analysis in this setting does not bring further knowledge to the results deduced from the individual papers, easily "attainable" without elevating the analysis to more complex levels.
In conclusion, for the degree of evidence requested by a drug already widely tested and applied and for the uniformity of the conclusive data on the topic contained in the individual (few) studies, I consider a more complex over-analysis in this topic to be of little use.
Meta-analyses should be reserved to controversial topics in which they can highlight or clarify the singularly unclear points by means of a wider statistical study weighting the single (different) evidences.
Author Response
Reviewer #2
The paper in question is a meta-analysis on the use of acetazolamide in heart failure patients, investigating its use both as a diuretic and as a sleep regulatory agent.
The study is well conducted from the procedural point of view. The technique used in the meta-analysis is correct, responds to widely accepted criteria and reaches well-documented conclusions.
However, I would like to highlight some critical points:
Response: We thank you for reviewing our manuscript and for your critical evaluation. We really appreciated your input and found your suggestions very helpful.
Comment#1 not all of the 9 articles included in the meta-analysis are cited in the bibliography; only two are mentioned, one of which, if I understand correctly, with the wrong year (2013 in the table, 2014 in references).
Response: We appreciated the reviewer’s important input and observation. The reviewer raised very important point. We apologize for these errors. We have made revisions as the reviewer’s suggestion. Table 1 year is now corrected to 2014. We have reviewed and corrected throughout revised manuscript as the reviewer’s suggestion.
Comment#2 Acetazolamide is an "old" drug, known since the 50s, which was widely used at the beginning, then it was replaced by loop diuretics and recently it is back in vogue as support / augmentation diuretic agent in cases of more refractory decompensation; data on the latter recent use are well known and accepted.
Response: We agree with the reviewer. We have observed clinical practice of using acetazolamide for diuresis but studies in heart failure population were lacking, therefore, we have not observed it being use as common as thiazide diuretics for diuresis augmentation. We speculated the potential electrolyte imbalance induced by acetazolamide might possibly be the concern. We appreciated the acknowledgement of the ongoing clinical trials looking at this specifically but unfortunately the result was not published yet. One clinical trial that reviewer 2 mentioned is ADVOR trial is now undergoing recruitment. Another clinical trial is investigating spironolactone, acetazolamide compare with placebo. We now included the sentence addressing the ongoing clinical trial in the introduction. We viewed that this systematic review will serve as a comprehensive way to extract all relevant prior data surrounding this question which may help serve a better understanding and interpretation of upcoming clinical trial.
Comment#3 It follows that the need for a meta-analysis is a bit "in the background" and this is even more true considering the low number of studies selected and included in the analysis (9 studies, 3 of them RCT, 2 on sleep disorder and just 1 on diuretic effect).
Response: The reviewer is very thorough and has made very good point. For the meta-analysis of the diuretic effect, we extracted the data from several observation studies. Although only one study, Verbrugge 2015, investigated patients with volume overload in modern medicine criteria, other observational studies included at least congestive heart failure patients with varying degree of excess volume. Studies from Fontana and Apostolo which were primarily geared toward sleep disorder did obtain urinary and serum chemistry which could also provide information in regards to the effect of acetazolamide in heart failure patients. Therefore, we viewed that pooling the values from these studies would serve as a good representation for the clinical spectrum of heart failure patients that received acetazolamide.
Comment#4 Authors themselves underline a limitation related to the cohort studies, so practically the meta-analysis on the diuretic effect is based just on a single paper.
Response: We appreciated the reviewer’s input. The reviewer made very good point. We performed an additional search of the literature to cover recently published articles up to the time of publication. We identified 7 other items (updated references# 34-40), 2 of which would qualify for our screening criteria. However, both of the articles did not contain our outcome of interest even though part of the studies involved the usage acetazolamide. We respect the reviewer and we have additionally emphasized this as the limitation of our stud that future large-scale clinical trials are required to confirm these effects of acetazolamide therapy on acid/base balance and apnea indexes.
Comment#5 The uniformity of the results of the studies is confirmed by the pooled effect estimate from the sensitivity analysis that remained essentially unchanged and by the absence of any publication bias
Response: The reviewer is correct and knowledgeable in meta-analysis. We are happy to remove this section if the reviewer and the editor think this should be removed. We performed the sensitivity analysis based on our planned analysis per protocol and we respect the reviewer and editor’s decision.
Comment#6 The study dates back a little more than a year ago (2017) but in 2018 another 6 studies were published, one of which was randomized (ADVOR, still recruiting): perhaps it was also worth including them in the analysis.
Response: We appreciated the reviewer’s thorough and helpful input. We agree with the reviewer and additionally performed the literature search. There are 6 clinical trials registered in clinicaltrials.gov. Only 1 study, Predicting Successful Sleep Apnea Treatment With Acetazolamide in Heart Failure Patients (HF-ACZ), has a preliminary result. It is a study with the primary outcome of sleep apnea index. Unfortunately, the primary outcome of the apnea-hypopnea index was not published, therefore, preventing us from incorporating into the analysis. This also pointed out the current topic of interest and importance of sleep disorder in this patient population, which this manuscript could provide a background review of the literature.
Of the literature published, we performed a crude PubMed search using acetazolamide and heart failure to ensure the capture of all publication. We also performed a search with our search term up extending to the current year. We identified 7 articles, consisting of 3 review articles, 1 case report, 1 editorial review, 1 factorial randomized controlled trial and 1 retrospective cohort study as shown in the table here. However, the 2 later articles did not contain our outcomes of interest. In Nunez et al., the study reported changes in NYHA class, weight, antigen carbohydrate 125 with the addition of acetazolamide to standard intensive diuretic strategy. Electrolytes changes were displayed in figure 2 but lacked the exact number that could be added into our analysis. Vergrugge et al. conducted a randomized control trial with 2x2 factorial design aiming to explore the effect of both acetazolamide and the use of spironolactone at discharge or upfront. The result section only highlighted finding on the comparison between the two strategies of spironolactone usage; thus, it could not be integrated into our analysis. In conjunction with your suggestion, we added a paragraph in the discussion section with a summary of update in these article up to the time of publication.
First author | Title | Journal | Year | Comment |
Terziyski K | Central Sleep Apnea with Cheyne-Stokes Breathing in Heart Failure - From Research to Clinical Practice and Beyond | Adv Exp Med Biol | 2018 | Review article |
Verbrugge FH | Spironolactone to increase natriuresis in congestive heart failure with cardiorenal syndrome. | Acta Cardiol | 2018 | Factorial RCT |
Núñez J | Use of acetazolamide in the treatment of patients with refractory congestive heart failure | Cardiovasc Ther | 2018 | Retrospective cohort |
Mullens W | Rationale and design of the ADVOR (Acetazolamide in Decompensated Heart Failure with Volume Overload) trial. | Eur J Heart Fail | 2018 | Review article |
Kataoka H | Treatment of hypochloremia with acetazolamide in an advanced heart failure patient and importance of monitoring urinary electrolytes. | Kataoka H | 2017 | Case report |
León Jiménez D | Diuretic treatment of the patient with diabetes and heart failure. Role of SGLT2 inhibitors and similarities with carbonic anhydrase inhibitors | Rev Clin Esp | 2018 | Article in Spanish, review article |
Verbrugge FH | Editor's Choice-Diuretic resistance in acute heart failure | Verbrugge FH | 2018 | Editorial article |
Comment#7 Meta-analysis is a statistical procedure for combining data from multiple studies. When the treatment effect is consistent from one study to the next, meta-analysis can be used to identify this common effect. When the effect varies from one study to the next, meta-analysis may be used to identify the reason for the variation. When the utility of an intervention or the validity of a hypothesis cannot be based on the results of a single study, meta-analysis is needed to synthesize data across studies. Narrative review is largely subjective and become unuseful when there are more than a few studies involved, mainly if the studies come to different conclusions. Meta-analysis, by contrast, applies objective formulas and can be used with any number of studies, mainly if the studies come to different conclusions.
There are examples of extensive meta-analyses that help a lot in the overall view of a problem, because it would be impossible for a single person to analyze all the articles related to it and draw general weighted conclusions.
(for example: Gong B, J Cardiothorac Vasc Anesth 2015 Dec 29(6) on levosimendan, collecting 5349 pts from 25 RCTs; Burnett H, Circulation Heart Failure 2017 Jan 13 on drug for HF collecting 57 RCTs studies).
In a case like this there is a low number of articles that all agree with each other, reporting, as confirmed by the Authors themselves, the same conclusions. In particular, the diuretic effect is analysed just by means of a single, already known RCT.
Therefore I believe that a meta-analysis in this setting does not bring further knowledge to the results deduced from the individual papers, easily "attainable" without elevating the analysis to more complex levels.
In conclusion, for the degree of evidence requested by a drug already widely tested and applied and for the uniformity of the conclusive data on the topic contained in the individual (few) studies, I consider a more complex over-analysis in this topic to be of little use.
Meta-analyses should be reserved to controversial topics in which they can highlight or clarify the singularly unclear points by means of a wider statistical study weighting the single (different) evidences.
Response: The reviewer is very thorough and we appreciate very kind and helpful comments to improve our manuscript. We agree with the reviewer’s input. The knowledge gap that we identified in the use of acetazolamide was in the electrolyte imbalance and pH changes. Although suggested by the drug mechanism, the effect in heart failure patient may be different when applied in the clinical setting. For example, improvement or worsening of sleep apnea may result in changes in serum bicarbonate level when acetazolamide was used. Therefore, we conducted the comprehensive review to ensure that there is a therapeutic effect and also that the evidence of serious adverse event were minimal to small. These findings hopefully may reassure the use of the medication in this condition. We respected the reviewer and have also additionally performed the literature search as the reviewer suggestions.
All authors thank the Editors and reviewers for their valuable suggestions. The manuscript has been improved considerably by the suggested revisions!

Reviewer 3 Report
In this meta-analysis manuscript, Wongboonsin et al. aimed to assess effects of acetazolamide therapy on acid/base balance and apnea indexes in patients with heart failure (HF). Overall, this is a straightforward, aim derived and concise meta-analysis study. The authors selected 9 previous published articles including either clinical trials or cohort studies with acetazolamide therapy in heart failure patients from a screening of 614 studies. The authors concluded that acetazolamide may have a potential beneficial effect in select heart failure patients by reducing serum bicarbonate level and augmenting natriuresis with only slight changes in serum pH and pCO2, and improvements in apnea indexes among HF patients. While the conclusion has considerable significance for clinical reference, this manuscript needs to be revised based on the following concerns:
1) A successful meta-analysis was based on the unbiased selection of all related studies. Although the authors described the search methodology as shown in Figure 1, they did not show the list of articles (at least list the information of the 43 candidates in Eligibility Step as an individual document, if not all) and thus it was not possible to assess the screen process.
2) There are only 2 cohort studies which is from the same group were included in apnea indexes analysis in this manuscript. Thus, this reviewer doubts the sample size for a meaningful meta-analysis. In addition, it cannot be excluded from bias since the author did not evaluate the publication bias in apnea indexes.
3) As the author mentioned, potential deleterious effects of acetazolamide were detected including metabolic acidosis, fatigability of respiratory muscles, nausea, vomiting, paresthesia, dizziness and muscle cramps. It was also notice that acetazolamide augments the hypercapnic ventilatory response in patients with heart failure (Javaheri 2014). However, the author did not analyze the deleterious effects with in the selected studies. At least, the author should list the reported side effect in the Table if meta-analysis could not be conducted because of small sample size.
4) Since there are many studies using acetazolamide in non-heart failure CSA patients [1-5], please discuss the common features and difference.
Minor issues:
1) Cite the 9 selected articles in the main text, figures or Tables.
2) In Figure 2A, the value of Variance is 0.000 in each study, and the P value was 0.000 in trial by Schwartz, 1954. Please indicate the reason in figure legend.
3) Since Figure S1 is an important data of the study according to the conclusion, why not show it in main text?
References:
[1] D.P. White, C.W. Zwillich, C.K. Pickett, N.J. Douglas, L.J. Findley, J.V. Weil, Central sleep apnea. Improvement with acetazolamide therapy, Arch Intern Med 142(10) (1982) 1816-9.
[2] E.T. Shore, R.P. Millman, Central sleep apnea and acetazolamide therapy, Arch Intern Med 143(6) (1983) 1278, 1280.
[3] H. Tojima, F. Kunitomo, H. Kimura, K. Tatsumi, T. Kuriyama, Y. Honda, Effects of acetazolamide in patients with the sleep apnoea syndrome, Thorax 43(2) (1988) 113-9.
[4] J. Verbraecken, M. Willemen, W. De Cock, E. Coen, P. Van de Heyning, W. De Backer, Central sleep apnea after interrupting longterm acetazolamide therapy, Respir Physiol 112(1) (1998) 59-70.
[5] R.N. Aurora, S. Chowdhuri, K. Ramar, S.R. Bista, K.R. Casey, C.I. Lamm, D.A. Kristo, J.M. Mallea, J.A. Rowley, R.S. Zak, S.L. Tracy, The treatment of central sleep apnea syndromes in adults: practice parameters with an evidence-based literature review and meta-analyses, Sleep 35(1) (2012) 17-40.
Author Response
Reviewer #3
In this meta-analysis manuscript, Wongboonsin et al. aimed to assess effects of acetazolamide therapy on acid/base balance and apnea indexes in patients with heart failure (HF). Overall, this is a straightforward, aim derived and concise meta-analysis study. The authors selected 9 previous published articles including either clinical trials or cohort studies with acetazolamide therapy in heart failure patients from a screening of 614 studies. The authors concluded that acetazolamide may have a potential beneficial effect in select heart failure patients by reducing serum bicarbonate level and augmenting natriuresis with only slight changes in serum pH and pCO2, and improvements in apnea indexes among HF patients. While the conclusion has considerable significance for clinical reference, this manuscript needs to be revised based on the following concerns:
Response: We thank you for reviewing our manuscript and for your critical evaluation. We really appreciated your input and found your suggestions very helpful.
Comment#1 A successful meta-analysis was based on the unbiased selection of all related studies. Although the authors described the search methodology as shown in Figure 1, they did not show the list of articles (at least list the information of the 43 candidates in Eligibility Step as an individual document, if not all) and thus it was not possible to assess the screen process.
Response: We appreciated the reviewer’s important comments. We agree with the reviewer and appreciated the reviewer’s comments to help improve transparency of our systematic review. Thus, we have added the list of 43 candidate articles in the supplementary data 1 following our search term. 4 were due to duplication of search. 9 articles were now included in the main article citation. The list in the supplementary data 2 displayed the 30 articles that underwent full review.
Comment#2 2) There are only 2 cohort studies which is from the same group were included in apnea indexes analysis in this manuscript. Thus, this reviewer doubts the sample size for a meaningful meta-analysis. In addition, it cannot be excluded from bias since the author did not evaluate the publication bias in apnea indexes.
Response: The reviewer is very thorough and has made very good point. We appreciated the reviewer input. The reviewer is correct about the publication bias for apnea indexes. Due to limited number of included studies evaluating the changes of AHI and CAI, the power of the test is too low to evaluate the publication bias. We respected the reviewer and we have added this point as the reviewer’s suggestion in the result as well as limitation of our study that future large-scale clinical trials are required to confirm these effects of acetazolamide therapy on acid/base balance and apnea indexes.
Comment#3 As the author mentioned, potential deleterious effects of acetazolamide were detected including metabolic acidosis, fatigability of respiratory muscles, nausea, vomiting, paresthesia, dizziness and muscle cramps. It was also notice that acetazolamide augments the hypercapnic ventilatory response in patients with heart failure (Javaheri 2014). However, the author did not analyze the deleterious effects with in the selected studies. At least, the author should list the reported side effect in the Table if meta-analysis could not be conducted because of small sample size.
Response: We appreciate the reviewer’s input. In the study Javaheri 2006, one patient developed shortness of breath and was thought to be due to induction of ventilator response. The author discussed that the dosage used in the study was much higher than in clinical practice, leading to a pronounced effect of the medication. None of the patients complained of paresthesia. We now added the comment about deleterious effects in the discussion as the reviewer’s suggestion.
Comment#4 Since there are many studies using acetazolamide in non-heart failure CSA patients [1-5], please discuss the common features and difference.
Response: We agree with the reviewer. We thanked the reviewer for pointing out this part for the discussion and we added this part into our discussion of the paper with all suggested references have been added. Aurora et al. published a comprehensive literature review on the treatment of central sleep apnea syndrome. The difference of CSA patients between non-heart failure and heart failure was highlighted given the section of recommendation was separated. There were at least 6 different forms of CSA based on the International Classification of Sleep Disorder (ICSD): 1) Primary CSA, 2) CSA due to Cheyne Stokes breathing pattern 3) CSA due to medical condition not Cheyne Stokes 4) CSA due to high-altitude periodic breathing 5) CSA due to drug or substance 6) Primary sleep apnea of infancy. Post-hypocapnia hyperventilation was believed to be the underlying pathobiology of CHF, high altitude sickness and primary CSA, although the patient with CHF could also elicit CSA with Cheyne Stokes breathing pattern. Most evidence of the use of acetazolamide stemmed from the study from high-altitude sickness population. White et al. was one of the foundation articles in primary CSA treatment with high-dose acetazolamide. The group showed a 69% improvement in total apneas after 1 week of therapy. Shore et al. reported a case that the use of acetazolamide resulted in a change of CSA to obstructive sleep apnea symptoms; therefore, proposed closed monitoring for a patient prescribed with acetazolamide for CSA. However, Tojima et al., conducted a case series that reported improvement of OSA by the use of acetazolamide. It appeared that the use of acetazolamide for CSA became a subject of interest, leading to Verbraecken et al., conducting an experimental study examining the effect of acetazolamide cessation in patients with CSA. The group concluded that acetazolamide could induce prolonged reset of CO2 threshold that may persist up to 6 months after acetazolamide was held. Given the complexity in the non-heart failure CSA patients, ongoing research exploring the effect of acetazolamide in CHF patient should consider this and may benefit from additional outcome adjudication after acetazolamide therapy was interrupted.
Minor issues:
1) Cite the 9 selected articles in the main text, figures or Tables.
Response: We agree with the reviewer. We have now added all 9 articles as references in our manuscript.
2) In Figure 2A, the value of Variance is 0.000 in each study, and the P value was 0.000 in trial by Schwartz, 1954. Please indicate the reason in figure legend.
Response: We apologize for being unclear. The actual variance is 0.0004 and P-value is 0.0002. We have now corrected Figure 2A as the reviewer’s suggestion.
3) Since Figure S1 is an important data of the study according to the conclusion, why not show it in main text?
Response: We agree with the reviewer. As the reviewer’s suggestion, we have now added the Figure S1 in the main manuscript to Figure 3 and change Figure 3 to Figure 4.
References:
[1] D.P. White, C.W. Zwillich, C.K. Pickett, N.J. Douglas, L.J. Findley, J.V. Weil, Central sleep apnea. Improvement with acetazolamide therapy, Arch Intern Med 142(10) (1982) 1816-9.
[2] E.T. Shore, R.P. Millman, Central sleep apnea and acetazolamide therapy, Arch Intern Med 143(6) (1983) 1278, 1280.
[3] H. Tojima, F. Kunitomo, H. Kimura, K. Tatsumi, T. Kuriyama, Y. Honda, Effects of acetazolamide in patients with the sleep apnoea syndrome, Thorax 43(2) (1988) 113-9.
[4] J. Verbraecken, M. Willemen, W. De Cock, E. Coen, P. Van de Heyning, W. De Backer, Central sleep apnea after interrupting longterm acetazolamide therapy, Respir Physiol 112(1) (1998) 59-70.
[5] R.N. Aurora, S. Chowdhuri, K. Ramar, S.R. Bista, K.R. Casey, C.I. Lamm, D.A. Kristo, J.M. Mallea, J.A. Rowley, R.S. Zak, S.L. Tracy, The treatment of central sleep apnea syndromes in adults: practice parameters with an evidence-based literature review and meta-analyses, Sleep 35(1) (2012) 17-40.
All authors thank the Editors and reviewers for their valuable suggestions. The manuscript has been improved considerably by the suggested revisions!

Round 2
Reviewer 2 Report
I appreciate the review carried out by the Authors and I understand very well the commitment behind it. Despite this, the meta-analysis remained limited to the few articles already presented in the first draft, so it remains a meta-analysis of a small number of articles. The conclusions are the same that can be deduced from the individual articles themselves. It follows that the usefulness of the present meta-analysis is extremely limited.
Author Response
I appreciate the review carried out by the Authors and I understand very well the commitment behind it. Despite this, the meta-analysis remained limited to the few articles already presented in the first draft, so it remains a meta-analysis of a small number of articles. The conclusions are the same that can be deduced from the individual articles themselves. It follows that the usefulness of the present meta-analysis is extremely limited.
Response: We greatly appreciated your very kind and helpful comments to improve our manuscript. We found your suggestions very helpful. As your suggestion, we performed an additional search of the literature to cover recently published articles up to the time of publication. We identified 7 other items (updated references# 34-40), 2 of which would qualify for our screening criteria. We respect the reviewer and we have additionally emphasized this as the limitation of our stud that future large-scale clinical trials are required to confirm these effects of acetazolamide therapy on acid/base balance and apnea indexes.
There are 6 clinical trials registered in clinicaltrials.gov. Only 1 study, Predicting Successful Sleep Apnea Treatment With Acetazolamide in Heart Failure Patients (HF-ACZ), has a preliminary result. It is a study with the primary outcome of sleep apnea index. We also additionally pointed out the current topic of interest and importance of sleep disorder in this patient population, which this manuscript could provide a background review of the literature.
The knowledge gap that we identified in the use of acetazolamide was in the electrolyte imbalance and pH changes. Although suggested by the drug mechanism, the effect in heart failure patient may be different when applied in the clinical setting. For example, improvement or worsening of sleep apnea may result in changes in serum bicarbonate level when acetazolamide was used. Therefore, we conducted the comprehensive review to ensure that there is a therapeutic effect and also that the evidence of serious adverse event were minimal to small. These findings hopefully may reassure the use of the medication in this condition.
The protocol for this systematic review is registered with PROSPERO (International Prospective Register of Systematic Reviews; no.CRD42017065401)
We respected the reviewer and have also additionally performed the literature search as the reviewer suggestions.
All authors thank the Editors and reviewers for their valuable suggestions. The manuscript has been improved considerably by the suggested revisions!

Reviewer 3 Report
The authors have answered or made necessary change according to my questions.
Minor:
1. In Figure 2A, the authors have updated the value of Variance and P value, please make the same modification to Figure 2C P value.
2. Line 245, the citation of 31 was duplicated.
Author Response
Reviewer #3
The authors have answered or made necessary change according to my questions.
Response: We thank you for reviewing our manuscript and for your critical evaluation. We really appreciated your input and found your suggestions very helpful.
Comment#1 In Figure 2A, the authors have updated the value of Variance and P value, please make the same modification to Figure 2C P value.
Response: The reviewer is very thorough and has made very good point. We agree and have made additional medication to Figure 2C P value as the reviewer’s suggestion.
Comment#2 Line 245, the citation of 31 was duplicated.
Response: We appreciate the reviewer’s input. We have removed the duplicated citation of 31 as the reviewer’s suggestion.
Once again, all authors thank the Editors and reviewers for their valuable suggestions. The manuscript has been improved considerably by the suggested revisions!
